# Large evanescently-induced Brillouin scattering at the surrounding of a nanofibre

Fan Yang [1,4 ✉], Flavien Gyger[1,5], Adrien Godet[2], Jacques Chrétien[2], Li Zhang[1], Meng Pang[3], Jean-Charles Beugnot [2 ✉] & Luc Thévenaz [1]

Brillouin scattering has been widely exploited for advanced photonics functionalities such as microwave photonics, signal processing, sensing, lasing, and more recently in micro- and nano-photonic waveguides. Most of the works have focused on the opto-acoustic interaction driven from the core region of micro- and nano-waveguides. Here we observe, for the first time, an efficient Brillouin scattering generated by an evanescent field nearby a single-pass sub-wavelength waveguide embedded in a pressurised gas cell, with a maximum gain coefficient of $18.90 \pm 0.17$ m$^{-1}$W$^{-1}$. This gain is 11 times larger than the highest Brillouin gain obtained in a hollow-core fibre and 79 times larger than in a standard single-mode fibre. The realisation of strong free-space Brillouin scattering from a waveguide benefits from the flexibility of confined light while providing a direct access to the opto-acoustic interaction, as required in free-space optoacoustics such as Brillouin spectroscopy and microscopy. Therefore, our work creates an important bridge between Brillouin scattering in waveguides, Brillouin spectroscopy and microscopy, and opens new avenues in light-sound interactions, optomechanics, sensing, lasing and imaging.

[1] Ecole Polytechnique Fédérale de Lausanne (EPFL), Group for Fibre Optics, CH-1015 Lausanne, Switzerland. [2] FEMTO-ST Institute, UMR 6174, Université Bourgogne Franche-Comté, 25030 Besançon, France. [3] State Key Laboratory of High Field Laser Physics, Shanghai Institute of Optics and Fine Mechanics, CAS, Shanghai 201800, China. [4] Present address: European Molecular Biology Laboratory, Heidelberg, Germany. [5] Present address: Max Planck Institute of Quantum Optics, Garching, Germany. ✉email: fanyang808@gmail.com; jean-charles.beugnot@femto-st.fr

**B**rillouin scattering involves light-sound interactions and has been used in nonlinear optics[1-4], microwave photonics[5], slow and fast light[6], lasing[7], sensing[8] and imaging[9,10]. It has been observed in various platforms, including optical fibres[8,11-14], whispering-gallery-mode resonators[15-18] and integrated waveguides[19-26].

The advent of micro- and nanophotonic waveguides has recently driven a renewed interest for Brillouin scattering as a tool to perform nonlinear optics and optical signal processing in waveguides. In 2006, a photonic crystal fibre was used to greatly intensify the Brillouin interactions by confining both acoustic and optical fields to a 1 µm microstructured core[12]. Another interesting work proposed that Brillouin interactions could be drastically enhanced by radiation pressure on subwavelength-scale waveguide boundaries[27], and triggered intense research interest into Brillouin scattering in micro- and nanophotonic waveguides. Since the gain of the interaction is proportional to the acousto-optic overlap integral, as well as to the pump power, two different approaches have been addressed to increase the Brillouin interactions: the first one focused on the design of waveguides with good optical and acoustic fields overlap[12,13,19-21], while the second approach used ultra-low-loss waveguides with high-Q resonant enhancement[15-18,23-25]. It should be mentioned that ultra-narrow linewidth Brillouin lasers based on evanescent Brillouin scattering using dilute optical mode and unguided leaky acoustic mode, have recently been demonstrated in micro-resonator waveguides with $Si_3N_4$ core and silica cladding[24,25]. The single-pass Brillouin gain coefficients are 0.1 $m^{-1}W^{-1}$ at 1550 nm[24] and 0.49 $m^{-1}W^{-1}$ at 674 nm[25], respectively, and the substantial Brillouin gains result from the high-Q resonant enhancement. So far, no significant Brillouin scattering has been generated from a single-pass optical waveguide into the surrounding medium located in the waveguide's vicinity.

Free-space Brillouin scattering has found applications in microscopy: in the past decade, Brillouin microscopy has been used for analysing the mechanical properties of biological samples[28] and hydrogel materials[29]. It provides label-free, non-contact, 3D imaging capabilities at typical optical resolution. So far, Brillouin microscopy has exclusively used free-space illumination, which requires careful optical alignment and has limited throughput[30]. Fluorescence microscopy based on total internal reflection in photonic waveguides[31] and structured illumination microscopy[32] solve these issues for the case of fluorescence microscopy; however, there is currently no alternative for Brillouin microscopy.

Here, we use a nanofibre waveguide and report the observation of strong single-pass Brillouin scattering generated by the evanescent field of a guided lightwave in pressurised carbon dioxide gas located in the immediate vicinity of the nanowaveguide. We observe drastic Brillouin scattering enhancement by increasing the gas pressure. We obtain a 11-times higher peak Brillouin gain coefficient in the nanofibre gas cell compared to the highest Brillouin gain realised in a hollow-core fibre gas cell[8] and 79-times higher compared to that in a standard single-mode fibre. Our Brillouin gain coefficient is 15-times higher compared to a small-core highly nonlinear microstructured fibre (1.23 $m^{-1}W^{-1}$)[33] and 8-times smaller compared to a high-index chalcogenide fibre (153.8 $m^{-1}W^{-1}$)[34]. We also measure the Brillouin spectra for different types of gases such as carbon dioxide ($CO_2$), sulphur hexafluoride ($SF_6$) and nitrogen ($N_2$). Our results pave the way for many potential applications in sensing, lasing, signal processing, etc and suggest a way to realise a novel waveguide-based Brillouin spectroscopy and microscopy that no longer requires free-space illumination.

## Results

**Operation scheme.** Brillouin scattering is an inelastic process within a medium involving a net energy transfer from the optical fields to the medium or vice versa. Spontaneous Brillouin scattering occurs when light scatters from thermally-excited sound waves, giving rise to frequency-shifted Stokes (i.e. phonon generation) and anti-Stokes waves (phonon annihilation). A conceptual illustration of Brillouin scattering around a nanowaveguide is shown in Fig. 1(a). In a waveguide, only photons scattered in the backward and forward directions are guided and hence detected. Here we focus on backward Brillouin scattering. The backward scattering efficiency is maximised when phase matching between all interacting waves is satisfied. In spontaneous Brillouin scattering, both Stokes and anti-Stokes scattering are favourable for measuring the Brillouin spectrum[35]. For the Stokes process, an incident pump photon of frequency $\omega_P$ is converted to a lower frequency Stokes photon of frequency $\omega_S$ through the scattering from the acoustic wave and thereby creating a phonon of angular frequency $\Omega$. The phase-matching condition requires $\omega_P = \omega_S + \Omega$ and $k_P = k_S + q$, where $k_P$, $k_S$ and $q$ are the wave vectors of the pump, Stokes, and phonon modes, respectively. For the anti-Stokes process, an incident pump photon is converted to a higher frequency anti-Stokes photon through the scattering from the acoustic wave and thereby annihilating a phonon, but this does not impact perceivably on the total number of phonons that are thermally activated regarding their low frequency. In this work, we arbitrarily use Stokes scattering to measure the Brillouin spectrum. In this case, the Brillouin frequency shift $\nu_B$, which is the frequency difference

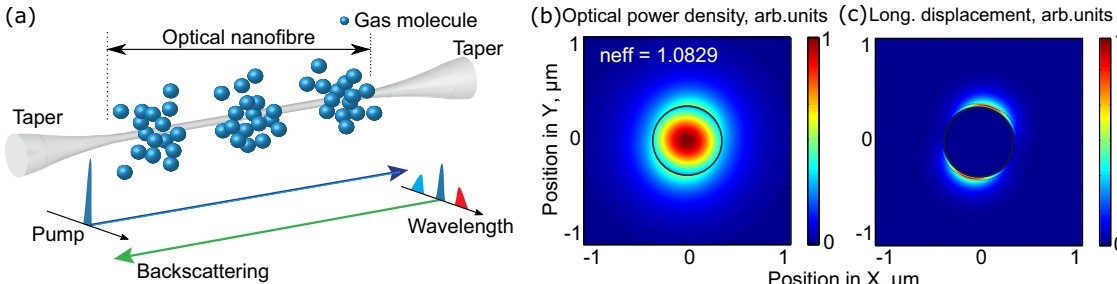

**Fig. 1 Brillouin scattering in a nanofibre gas cell. a** Conceptual view of the Brillouin scattering in a nanofibre gas cell. Pump (in green) goes from left to right and the backscattered light includes Rayleigh scattering from the same frequency and frequency down-shifted Stokes (in red) and frequency up-shifted anti-Stokes (in blue) Brillouin scattering. **b** Computed normalised spatial optical power distribution of the fundamental optical mode at a wavelength of 1550 nm for a 740 nm diameter nanofibre surrounded with 40 bar $CO_2$ gas. The effective mode refractive index is 1.0829. **c** Computed normalised longitudinal displacement of the elastic wave at 350 MHz as a result of backward Brillouin scattering in the surrounding gas. The black circles in (**b**) and (**c**) show the silica nanofibre boundary. See Methods for the simulation details.

between pump and Stokes waves under phase-matching condition, is given by[36]:

$$\nu_{\mathrm{B}} = 2n_{\mathrm{eff}} v_{\mathrm{a}}/\lambda, \qquad (1)$$

where $n_{\mathrm{eff}}$ is the effective refractive index of the optical mode, $v_{\mathrm{a}}$ is the acoustic velocity of the medium - the surrounding gas in our case - and $\lambda$ is the pump wavelength in vacuum.

The small nanofibre dimensions (740 nm in this work) compared to the optical wavelength results in 58% of the light field intensity propagating outside the nanofibre, which can interact with the surrounding gas or other fluid material. This evanescent field is visible in the finite-element simulation of the power distribution of the fundamental optical mode illustrated in Fig. 1(b). The gas molecular displacement of the acoustic mode along the fibre axis direction is shown in Fig. 1(c) (see Methods for the simulation details). The spatial distribution of the transverse displacement and the elastic energy density are shown in Supplementary Note 1.

**Nanofibre gas cell.** The Brillouin backscattering spectrum from an elastic wave generated in the gas surrounding the silica nanofibre can be neatly observed. The large evanescent field creates a strong electrostrictive force spatially distributed around the nanofibre. The elastic wave in the gas generated via electrostriction is confined by the evanescent optical field. Figure 2 shows a numerical calculation giving the map of the Brillouin spectrum in 40 bar $CO_2$ surrounding a nanofibre with diameters ranging from 0.5 to 1.5 μm. This calculation indicates the geometrical sizes of the nanofibre maximising the light-sound interaction in the surrounding gas, which is a trade-off between the optical mode's effective overlap and the optical intensity in the evanescent field. The optimal diameter of the nanofibre maximising Brillouin scattering by the evanescent field is calculated to be 740 nm.

The detailed fabrication process of the nanofibre gas cell is described in Methods. We infer the waist diameter of the nanofibre with a few nanometre sensitivity by mapping the backscattered Brillouin spectrum along the optical fibre taper and fitting with numerical simulations of the elastodynamic equations (a technique we developed in 2017[37]). The fabricated nanofibre sample presented in this work shows a waist diameter of 740 nm ± 5 nm, a waist length of 10 cm, two adiabatic transitions of 79.4 mm ± 5 mm of total length and an insertion loss of 0.14 dB. The backscattered Brillouin signal of the nanofibre is shown in Fig. 3. The blue and dotted red lines in Fig. 3 are respectively the experimental and numerical simulation results in the nanofibre after the tapering process. The experimental Brillouin spectrum, including the surface acoustic modes and hybrid acoustic modes, matches perfectly with the numerical simulation for a nanofibre with a diameter of 740 nm ± 5 nm. After fabrication, the nanofibre is packaged in a simple metallic tube as a gas cell while applying a controlled strain. The measured Brillouin spectrum after applying the strain is shown in the dotted black line of Fig. 3. The amplitude of the measured hybrid acoustic mode of the nanofibre with a strain of 2.15% at 9 GHz is 48. The measured resonant Brillouin frequency of the same hybrid acoustic mode before applying a strain is at 8.2 GHz and the amplitude is 63. The resonant Brillouin frequency of the simulation result before applying strain is at 8.2 GHz with an amplitude of 38. As a result of the strain, the resonances due to surface acoustic modes and hybrid acoustic modes both shift to higher frequencies, perfectly matching with our previous analysis[38].

**Evanescently-induced Brillouin scattering in the nanofibre gas cell.** The Brillouin gain coefficient and spectrum can be obtained from a spontaneous Brillouin scattering measurement[35]. The detailed calculation and calibration of the Brillouin gain coefficient from the spontaneous Brillouin measurements are described in Supplementary Note 2. All the experiments were performed at an environmental temperature of 24 ± 1°C. Figure 4(a) shows the detailed experimental implementation. The pump light is amplified by an erbium-doped fibre amplifier and launched through a circulator into the nanofibre gas cell. A 53.5 m single-mode fibre (SMF) is appended to the nanofibre and provides a direct comparative response for the Brillouin gain. The backward spontaneous Brillouin scattering signal is measured using heterodyne beating with a frequency-upshifted fraction of the pump laser (as a local oscillator) and detected by a radio-frequency electrical spectrum analyser via a photodetector. The beating spectra from the 40 bar $CO_2$ contribution in the 10 cm nanofibre gas cell, from the silica contribution in the 10 cm nanofibre as well as from the 53.5 m SMF are shown in Fig. 4(b). The Brillouin frequency shift for the 40 bar $CO_2$ contribution, silica contribution and SMF contribution are 340 MHz, 9 GHz and 10.86 GHz, respectively. Note that the 9 GHz signal is from the silica nanofibre hybrid acoustic mode which has been analysed in Fig. 3. The +110 MHz frequency-shifted local oscillator splits the Stokes and anti-Stokes components. The right peaks identify the Stokes scattering while the left peaks originate from the anti-Stokes scattering. Remarkably, we can observe that the peak Brillouin signal for 10 cm nanofibre filled with 40 bar $CO_2$ is ~10 times higher than that of the 53.5 m SMF.

The red and blue lines in the inset of Fig. 4(b) show the Brillouin gain spectra of the 40 bar $CO_2$ nanofibre gas cell and the SMF respectively, when the polarisations are tuned to maximise the specific Brillouin signal. The peak Brillouin gain coefficient for the nanofibre gas cell filled with 40 bar $CO_2$ is $8.20 ± 0.08 \, \mathrm{m}^{-1}\mathrm{W}^{-1}$ which is 34 times higher than that of the SMF ($0.24 ± 0.0022 \, \mathrm{m}^{-1}\mathrm{W}^{-1}$). The error bar is calculated from the standard deviation of the peak gain over 10 measurements. Note that the peak Brillouin gain coefficient for the SMF obtained from our spontaneous scattering measurement is in good agreement with the standard value for this type of fibre[39], which confirms the solidity of our test bench and calibrations based on spontaneous scattering. The shape of the Brillouin gain spectrum of the standard SMF is a fully symmetric Lorentzian distribution while the shape of the Brillouin gain spectrum of the 40 bar $CO_2$ nanofibre gas cell is evidently asymmetric, as explained in the next section.

**Different gas pressures.** Both Stokes and anti-Stokes Brillouin scattering have exactly the same spectrum. Figure 5(a) shows the Brillouin gain as a function of the $CO_2$ pressure in the nanofibre gas cell. The peak gain coefficient increases with the pressure while the linewidth is simultaneously reduced, in agreement with

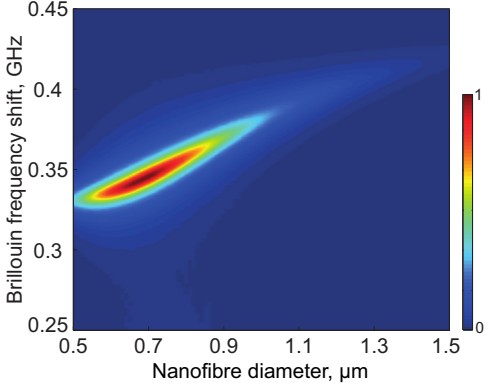

**Fig. 2 Numerical calculation of the Brillouin spectrum.** 3D calculation of normalised Brillouin scattering spectra in silica nanofibres surrounded by 40 bar $CO_2$ for a diameter ranging from 500 nm to 1.5 μm.

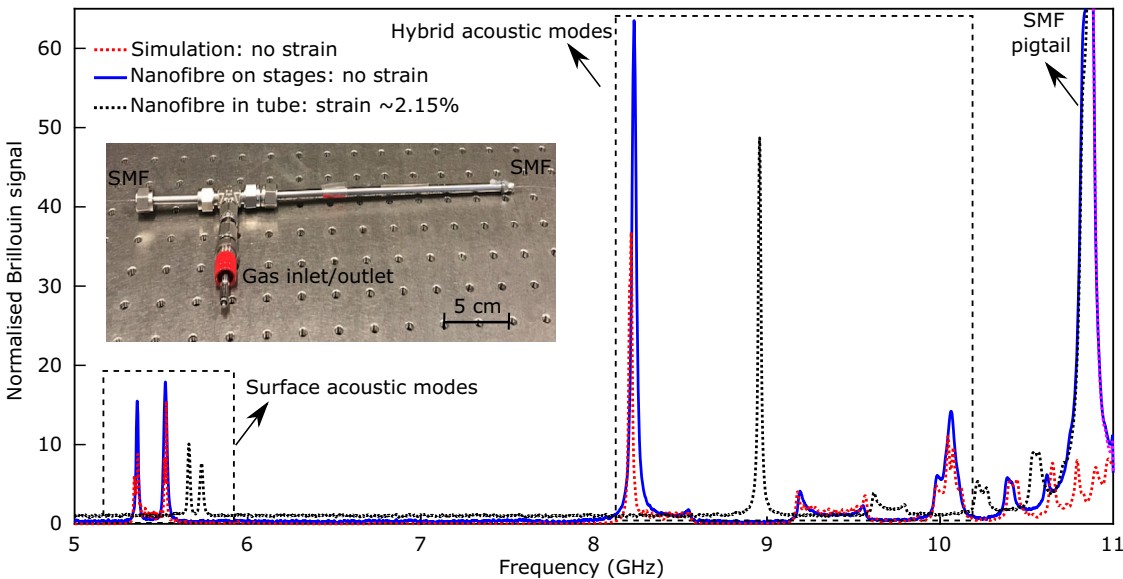

**Fig. 3 Backward Brillouin scattering in the silica material of the nanofibre during fabrication process.** The backward Brillouin scattering of the nanofibre is measured with a set-up we developed in 2017[37]. The Brillouin signal is normalised to the background noise of the electrical spectrum analyser when there is no pump. The Brillouin spectra from the surface acoustic modes and the hybrid acoustic modes of the silica nanofibre during fabrication are calculated by the model in[37]. The blue line shows the measured Brillouin spectrum of the nanofibre on the stages after tapering. The dotted red line shows the simulated Brillouin spectrum of the nanofibre with a waist diameter of 740 nm. The dotted black line shows the measured Brillouin spectrum of the nanofibre after packaging the nanofibre in a metallic tube with a 2.15% strain. Note that the signal from SMF pigtail is not taken into account in the simulation and any response from interactions in the gas is not visible in the frequency range covered by the set-up. The Brillouin scattering from the standard single-mode fibre pigtail of the nanofibre is also observed at a frequency of 10.86 GHz. The inset shows the fabricated nanofibre gas cell with the untapered SMF segments exiting the gas cell at each end. Gases can be pressurised in or vacuum pumped out through the gas inlet/outlet port.

the model presented in[8]. The peak gain coefficient is $18.90 \pm 0.17\,\mathrm{m^{-1}W^{-1}}$ at 57 bar which is 79 times stronger than that in a SMF. The error bar is calculated from the standard deviation of the peak gain for 10 measurements.

It is interesting to see in Fig. 5(a) that the shape of the Brillouin spectra is skewed. This asymmetry results from the contribution of the tapered transition region, since this segment shows a position-varying evanescent optical field that keeps interacting with the sound wave in the gas. Let us take 40 bar $CO_2$ as an example and analyse its spectrum in Fig. 5(b). The blue line is the theoretical Brillouin spectrum in 40 bar $CO_2$ (along the homogeneous nanofibre section) and the red line is the theoretical Brillouin spectrum including the uniform nanofibre section as well as the tapered region (see Supplementary Note 3 for the detailed analysis). The tapered region only contributes to the higher frequency region because its gradually larger diameter increases the effective refractive index, leading to a larger Brillouin frequency shift, as indicated by the phase-matching condition given in Eq. (1). So the tapered region presents a higher Brillouin frequency shift and makes the Brillouin spectrum of the nanofibre slightly skewed. The theoretical Brillouin spectrum taking into account this asymmetry matches well with our experimental results.

Figure 5(c) shows the measured Brillouin linewidth for the nanofibre gas cell filled with $CO_2$ at different pressures as well as the theoretical estimation of the linewidth due to the material damping. Here, to exclude the contribution from the tapered regions, the Brillouin linewidth for the nanofibre gas cell is defined as twice the value of the left half-width-half-maximum of the spectrum fitted with a Lorentzian (i.e. considering only the left half of the spectrum). The measured Brillouin linewidth decreases with pressure in the range from 5 to 50 bar. It then increases from 50 to 57 bar, as a result of the close vicinity of the gas-liquid phase transition at room temperature, since the acoustic damping increases when the gas phase is approaching

the liquid transition. It should be mentioned that the measured Brillouin linewidth for the nanofibre at a specific pressure is ~10 MHz larger than the linewidth induced by the material damping. This linewidth broadening is thought to originate from the phonon leakage due to the acoustic unguiding nature of the acoustic wave[40] as well as the coupling of the light field with a continuum of free-space modes[41]. The broadening effect has also been observed in acoustically unguiding solid optical fibre with a pure silica core and a fluorine-doped cladding[42].

**Different types of gas.** We then carried out the study of Brillouin scattering in a nanofibre gas cell filled with different types of gas, namely $CO_2$, sulphur hexafluoride ($SF_6$) and nitrogen ($N_2$). The Brillouin gain spectra for these three gases at specific pressures are shown in Fig. 6. The peak Brillouin coefficients for 2 bar $SF_6$, 5 bar $CO_2$ and 10 bar $N_2$ are 0.13 $\mathrm{m^{-1}W^{-1}}$, 0.34 $\mathrm{m^{-1}W^{-1}}$ and 0.32 $\mathrm{m^{-1}W^{-1}}$, respectively. The noise defined by the standard deviation of the signal off-peak of the Brillouin spectrum for 2 bar $SF_6$, 5 bar $CO_2$ and 10 bar $N_2$ are 0.0046 $\mathrm{m^{-1}W^{-1}}$, 0.013 $\mathrm{m^{-1}W^{-1}}$ and 0.015 $\mathrm{m^{-1}W^{-1}}$ respectively. The signal-to-noise ratio for 2 bar $SF_6$, 5 bar $CO_2$ and 10 bar $N_2$ are 28, 26 and 21 respectively. The results clearly show the possibilities of the evanescent Brillouin scattering of our nanofibre to be used for Brillouin spectroscopy and gas analysis, even at or close to ambient conditions, as well as the flexibility of our platform in tailoring the Brillouin gain at will.

## Discussion

In this work, we have demonstrated the generation of a strong Brillouin scattering driven by the evanescent field of a nanofibre placed in a gas-filled cell. The nanofibre is obtained by fused-tapering of a standard single-mode fibre, so that the mode size transition to a sub-micron dimension can be adiabatically realised with very low added loss: this is a crucial advantage for the implementation of such

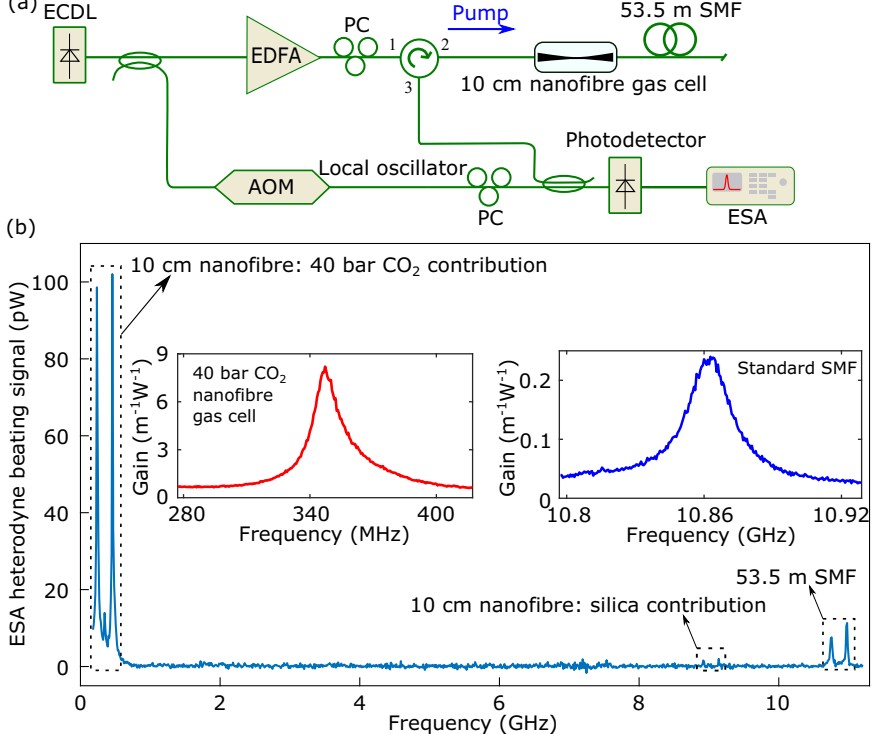

**Fig. 4 Backward Brillouin scattering in the nanofibre gas cell. a** Heterodyne experimental set-up. To make a comparison between peak Brillouin gains, a 53.5 m single-mode fibre (SMF) is connected to the far end of the nanofibre. The output of a continuous-wave external-cavity diode laser (ECDL) is split: one branch is amplified by an erbium-doped fibre amplifier (EDFA) and used to pump the nanofibre as well as the SMF. The other branch is up-shifted in frequency (+110 MHz) by an acousto-optic modulator (AOM) and combined with the backward spontaneous Brillouin scattering for heterodyne mixing. The total length of the nanofibre (including the adiabatic tapering transition and the nanofibre waist) is 179.4 mm ± 5 mm with a 10 cm long 740 nm ± 5 nm diameter nanofibre. Note that the total insertion loss of the nanofibre is 1 dB including the 0.14 dB tapering-induced loss, the splicing loss of the nanofibre to two fibre pigtails at each end, as well as the loss of the two angled-physical contact connectors. The heterodyne spectrum is measured by a photodetector combined with a radio-frequency electrical spectrum analyser (ESA). The AOM is used to separate the Stokes and anti-Stokes Brillouin scatterings. The polarisations of the pump light and the local oscillator are adjusted using two polarisation controllers (PCs) to achieve the highest heterodyne response. **b** Heterodyne beating spectra of the 10 cm nanofibre gas cell and the appended 53.5 m SMF. The Brillouin frequency shifts of the 40 bar $CO_2$ in the 10 cm nanofibre gas cell, the silica contribution in the 10 cm nanofibre gas cell as well as the 53.5 m SMF are 0.34 GHz, 9 GHz and 10.86 GHz, respectively. The red and blue lines in the inset of **(b)** are the backward Stokes Brillouin gain spectra for the 40 bar $CO_2$ nanofibre gas cell and for the SMF, respectively. The horizontal scales span over the same frequency width for a good comparison.

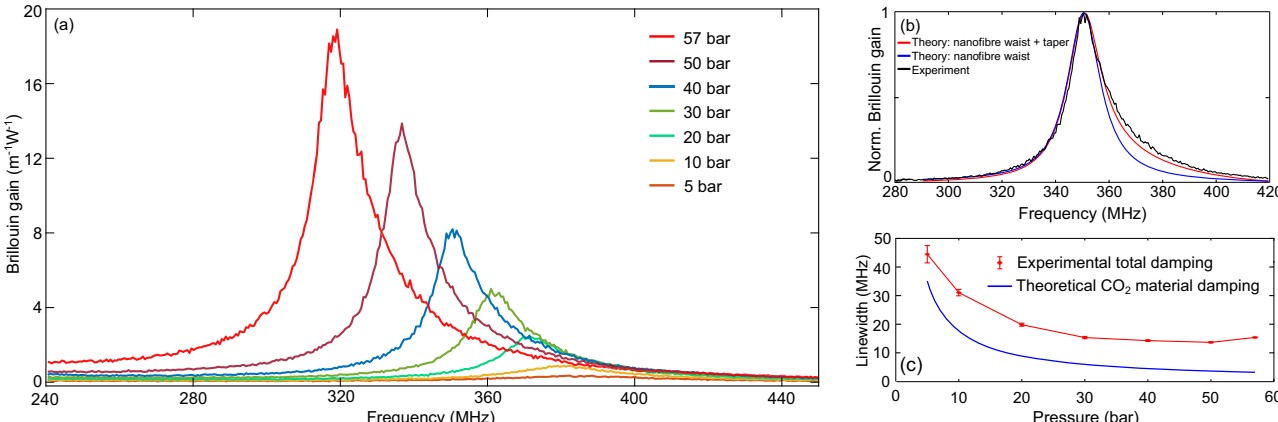

**Fig. 5 Experimental Brillouin gain spectra in the nanofibre placed in a gas cell filled with $CO_2$. a** Measured Brillouin gain spectra along the nanofibre gas cell filled with $CO_2$ at different pressures. **b** Normalised Brillouin gain spectra in 40 bar $CO_2$ obtained in a nanofibre with a waist of 740 nm. The blue line shows the theoretical Brillouin spectrum at the nanofibre waist; the red line shows the compound theoretical Brillouin spectrum along the nanofibre waist and the tapered region. The black line shows the experimental Brillouin spectrum along the nanofibre in the gas cell filled with 40 bar $CO_2$. **c** Measured Brillouin linewidth defined as twice the value of the left half-width measured spectrum using a Lorentzian fitting on the experimental curves in (**a**). Each red data point shows the estimated mean and the RMS error bar for a total of 10 measurements at each pressure. The blue line shows the theoretical Brillouin linewidth of free-space $CO_2$ gas at different pressures.

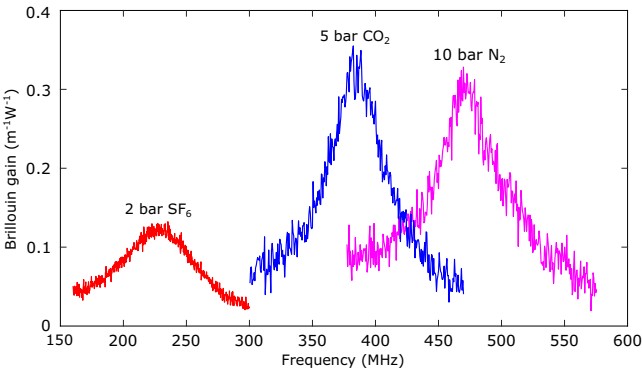

**Fig. 6 Experimental Brillouin gain spectra from the nanofibre gas cell filled with different types of gas.** Measured Brillouin gain spectra for 2 bar $SF_6$, 5 bar $CO_2$ and 10 bar $N_2$. For the $CO_2$ and $N_2$ measurements, the noise equivalent bandwidth of the electrical spectrum analyser (ESA) is 1 kHz, while for the $SF_6$ measurement, the noise equivalent bandwidth of the ESA is 100 Hz.

a gain stage in photonic systems. The tighter light confinement in nanofibre compared with hollow-core fibre results in a measured Brillouin gain coefficient in our 40 bar $CO_2$ gas filled nanofibre gas cell 5-times larger than in a hollow-core fibre under the same pressure[8], but it is expected that higher gains can be achieved by using noble gases (e.g. xenon)[43]. Using our nanofibre, we can increase the pressure close to the gas-liquid phase transition of $CO_2$ (64 bar at ambient temperature) with negligible absorption loss because of the relatively shorter length of the nanofibre and smaller percentage of light interact with the gas molecules, in contrast with hollow-core fibres where light molecular absorption turns out to limit the maximum pressure. In these conditions, we have achieved a peak Brillouin gain coefficient of $18.90 \pm 0.17$ m$^{-1}$W$^{-1}$ at 57 bar, which is 15 times higher than that obtained in a small-core highly nonlinear microstructured fibre and is 8 times smaller than that in a high-index chalcogenide fibre. It should be mentioned that the Brillouin gain coefficient of 57 bar $CO_2$ in units of m/W at 1550 nm is estimated to be $2.6 \times 10^{-10}$ m/W. Furthermore, unlike in hollow-core fibres[8] where both optical and acoustic modes are confined in the hollow core, this work has demonstrated the first possibility to probe the acoustic wave in the surrounding medium of a single-pass waveguide using an all-optical method thanks to the strong Brillouin scattering generated by the evanescent field of the waveguide.

This feature offers the attractive possibility to realise compact optical amplification stages, since a gain similar to that generated over several metres of standard fibre can be obtained over a few centimetres along the nanofibre, with a very minor insertion loss. The Brillouin gain coefficient we demonstrate in a nanofibre gas cell is two orders of magnitude smaller compared with that of a silicon waveguide[44]. The concept may be extended to on-chip waveguiding structure showing a substantial evanescent field and can be the essence of gain blocks in photonic integrated circuits. For instance, a suspended[21] or slot nano-waveguide[45] could be used for strong light-sound interactions in gas. Furthermore, our scheme shows that we can tailor the Brillouin gain spectrum by changing the gas pressure or using different type of gases. Various functional optical devices and sensors could be generated based on this platform. It should be mentioned that we use backward Brillouin scattering and the pump and signal light counter-propagate. Therefore, in principle, the proximity of the pump and signal does not pose issues for light amplification unless there is reflection point. The gain spectrum bandwidth of a nanofibre gas cell filled with 57 bar $CO_2$ is 15 MHz which is 2-fold narrower than a standard single-mode fibre. The peak gain coefficient of the nanofibre gas cell is 79-fold stronger than the single-mode fibre. We can modulate and engineer the pump light

spectrum to broaden the signal amplification spectrum because the amplification spectrum is the convolution of the pump light spectrum and the Brillouin gain spectrum.

Our nanofibre gas cell can also be used for novel approaches in pressure or temperature sensing. For instance, the measured Brillouin frequency shift along the nanofibre gas cell filled with $CO_2$ at different pressures (shown in Supplementary Fig. 6 in Supplementary Note 4) shows a pressure sensitivity of $\sim -1$ MHz/bar. This large sensitivity enables our nanofibre gas cell to be used for pressure measurement over a range from 5 to 57 bar. Temperature variations also change the gas acoustic velocity[8] and hence the Brillouin frequency shift in the nanofibre gas cell.

Micro and nanofibres with evanescent field have been used for chemical and biological sensing[46] based on absorption[47], fluorescence[48], elastic scattering[49] and Raman scattering[50]. So far, no waveguide-based evanescent Brillouin sensing has been reported. Brillouin scattering is used for measuring the mechanical properties of a material which absorption, fluorescence, elastic scattering and Raman scattering can not. The sound wave velocity and the material viscosity can be obtained from the Brillouin resonant frequency and the Brillouin linewidth of the Brillouin spectrum, respectively[51,52]. Our work provides an efficient way for Brillouin sensing and does not bring additional complexity compared with Raman or fluorescence-based sensing.

In order to estimate the capability of our nanofibre waveguide for microscopy and spectroscopy of biological samples, we have simulated the Brillouin gain spectrum of a nanofibre immersed in water. Since the acoustic velocities in the surrounding medium (e.g. $\sim 300$ m/s in gas and 1500 m/s in water) and that of the solid silica nanofibre (6000 m/s) are very different, the respective Brillouin frequency shifts in the surrounding medium and the nanofibre are well separated. The peak Brillouin gain is calculated to be 1 m$^{-1}$W$^{-1}$ for a 450 nm diameter nanofibre immersed in water, which is 8 times larger than that in 2 bar of $SF_6$ shown in Fig. 6 and 4 times larger than that of a standard single-mode fibre. The detailed simulation is described in Supplementary Fig. 7 in Supplementary Note 5. The accessibility of the acoustic wave, present in the surrounding medium and thus external to the waveguide, and the high Brillouin gain of a nanofibre immersed in water opens new perspectives and creates an important bridge between Brillouin scattering in waveguide and Brillouin microscopy and spectroscopy. So far, Brillouin microscopy and spectroscopy have exclusively relied on free-space light excitation. For Brillouin spectroscopy[29], our platform provides an efficient way for light illumination and Brillouin scattering signal collection. For Brillouin microscopy, our work suggests a novel imaging modality, namely waveguide-illumination Brillouin microscopy, which inherits many benefits of on-chip-based microscopy[31,32], such as separation between the illumination and detection light paths, facilitated alignment and high throughput[30].

## Methods
**Simulations**. The transverse dimension of the silica nanofibre is close to the acoustic wavelength. Therefore, in such a waveguide, the boundaries conditions induce a strong coupling between longitudinal, shear and surface elastic components. The generation of a confined elastic wave in the gas-silica rod assembling by the two components of optical field (guided and evanescent) is calculated by using the elasto-dynamic equation driven by the electrostrictive stress[53,54]. All calculations for $CO_2$ at 40 bar are realised using: an optical wavelength of 1550 nm, a refractive index of 1.01804[8], an acoustic velocity of 250 m/s, and a density of 71.35 kg/m$^3$. To exclude the contribution from the tapered regions, the Brillouin linewidth used in the simulation for 40 bar $CO_2$ is 14.2 MHz which is defined as twice the value of the left half-width-half-maximum of the spectrum fitted with a Lorentzian (i.e. considering only the left half of the spectrum).

**Fabrication of the nanofibre gas cell**. The nanofibre is fabricated by tapering a standard single-mode fibre (SMF) using the heat-brush technique. To package the nanofibre into the gas cell, we applied the following procedure: first, we move the nanofibre sample from the tapering translation stage to a fixed-length stage with

two fibre clamps on both ends of the SMF pigtails. Second, we orient the stage perpendicular to the ground, release the lower fibre clamp, insert the lower end of the nanofibre sample into a metallic tube, to finally hermetically glue the lower end of the tube. Third, to avoid any contact of the nanofibre with the tube wall, we apply a 2.15% strain on the nanofibre. We then apply glue on the other side while precisely controlling the strain and monitoring the backward Brillouin spectrum.

## Data availability

The data and code used to produce the plots within this paper are available at Zenodo (https://doi.org/10.5281/zenodo.5731429). All other data used in this study are available from the corresponding authors upon reasonable request.

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

## Acknowledgements

The authors would like to acknowledge the support by the Swiss National Science Foundation (SNSF) under grant No. 159897 and 178895, and the financial support of Agence Nationale de la Recherche (ANR-16-CE24-0010-03), EIPHI Graduate School (contract ANR-17-EURE-0002) and Bourgogne-Franche-Comté Region.

## Author contributions

F.Y. conceived the project. A.G. and J.C. fabricated and characterised the nanofibre gas cells. F.Y., F.G. and L.Z. performed the gas Brillouin experiments. J.-C.B. developed the

numerical model. F.Y., F.G., M.P. and J.-C.B. analysed the data. F.Y., F.G. and J.-C.B. wrote the manuscript with inputs from L.T. L.T. and J.-C.B. supervised the project.

## Competing interests

The authors declare no competing interests.
