## [Peer Review File · Nature Communications]

Large evanescently-induced Brillouin scattering at the surrounding of a nanofibreREVIEWER COMMENTS

Reviewer #1 (Remarks to the Author):

Authors present a paper entitled “Large evanescently-induced Brillouin scattering at the surrounding of a nanofibre” in which the evanescent field in a tapered fibre is used to excite Brillouin scattering in a gas surrounding the waveguide. The manuscript is well-written and relatively easy to follow. It is definitely interesting. I would recommend publication after Authors consider the following:

1) This method is very much in the spirit of other evanescent wave sensors. Tapered fibres (or even those that have gone through the painful process of polishing into a D shape) and their resulting evanescent fields have been used for chemical, biological, etc. sensing using a wide range of modalities. How does the additional complexity of a Brillouin apparatus compare with some of those other methods? Can Authors propose advantages to such a sensor in using Brillouin scattering rather than some other framework?

2) Authors state that “So far, no strong Brillouin scattering has been generated from an optical waveguide into the surrounding medium located in the waveguide’s vicinity.” The word “strong” is somewhat subjective and ambiguous. Can Authors provide examples where the evanescent (optical) wave was used to excite Brillouin scattering?

3) Authors consistently compare their results against those of a standard SMF, which has a very large mode size. A better comparison would be with fibres made from high-gain materials, such as high-index chalcogenides, or a microstructured highly nonlinear fibre. Looking at Fig. 4, even the “silica contribution” from the tapered fibre appears to have a higher gain coefficient than the conventional SMF.

Along those lines, is it possible to deduce a material value in units of m/W ? The values for the gain coefficient provided are dependent upon the area of the optical mode. It would be helpful to the reader in understanding this in the context of the tradeoff between mode intensity and overlap with the outside region. This, too, is a place where information is lacking for the water simulation.

4) As far as applications go, Authors state that “This feature offers the attractive possibility to realise compact optical amplification stages, since a gain similar to that generated over several tens of meter of standard fibre can be obtained over a few centimetres along the nanofibre, with a very minor insertion loss” and go on to state that this can be useful in photonic integrated circuits. There have been recent demonstrations of Brillouin amplification in, for example, silicon nanowaveguides where the high index gives rise to an even larger Brillouin gain. Can the proposed configuration outperform these other approaches? Even if so, do the proximity of the Brillouin frequency to the pump and the narrow width of the gain spectrum pose issues or obstacles for this application?

5) What was the purpose of adding the other gases since no numbers are given? Is a lower gain the cause of the noisy signal (I am assuming since the 5bar CO_2 case is shown, that is meant to be a comparison to a low-gain case)? What is the significance of the noise bandwidth?

6) Section S2: This discussion is somewhat on the light side and written as an afterthought. More detail would be useful. It would also be informative/illustrative to show a few Brillouin spectra together from different positions along the taper. I would have imagined that the contributions from the tapered regions, particularly their tails, would have impacted the red side of the spectra, as well as the blue, but not quite as much. Therefore, I wonder about the fittings used for the spectral width (half width on the left side, as Authors state, such as the blue curve in Fig. 5(b)). Perhaps this method may be overestimating the spectral width?

Reviewer #2 (Remarks to the Author):

In the manuscript, Yang et al. report Brillouin scattering in the evanescent field of a sub-wavelength waveguide in a high-pressure gas cell. The experiment results show clear Brillouin scattering from the evanescent field and are in good agreement with the simulation results. Results using different gas pressures and different gas species are compared to show the potential for Brillouin spectroscopy. The manuscript is well written.

However, one of the main claims the authors made is that the Brillouin gain coefficient is much larger than in their previous results and SM fiber in particular. While these statements are correct, it does not seem a fair comparison, as the current gas pressure used is 40 bar which is a very high number. Also, as stated by the authors, this pressure is already approaching the gas-liquid phase transition. For some of the lower pressures measured (e.g., the 5 bar trace in Fig. 5) the Brillouin scattering peak is hardly resolved. Even though the results are interesting and clearly presented, the usefulness and generality of the approach as a spectroscopic technique or new platform for various studies as mentioned in the abstract is in question. It seems that the results are more appropriately presented in a journal having a focused subject area.

The followings are some detailed points:

1. In Fig. 1a, the authors should provide discussion as to why Stokes and anti-Stokes are both favorable in the experiment?
2. In Fig. 1c, it seems like the authors want to demonstrate the acoustic field in the Brillouin process. Is this elastic wave entirely composed of CO₂, and confined to the waveguide boundary or is it a hybrid wave?
3. In Fig. 3, there are surface acoustic modes at 5-6 GHz (about three times smaller than 4. GHz). Can this feature still be resolved in Fig. 4?
5. In Fig. 4, the authors state that "The right peaks identify the Stokes scattering while the left peaks originate from the anti-Stokes scattering." As I understand, AOM will down-shift the LO frequency. Thus the left peak should also come from the down-shifted wave, which is the Stokes wave.
6. In Page 5, the gain coefficient for SMF should also be given with error bar.
7. In Fig. 6, why do the different lines have different RBW?

Reviewer #3 (Remarks to the Author):

In this paper, Fan et al. claim to demonstrate strong Brillouin scattering generated by the evanescent field of a sub-wavelength nanofiber waveguide in a pressurized gas cell. Compared with the authors' previous work of large Brillouin amplification in a gas-filled hollow-core-fiber at high pressure, the maximum gain coefficient achieved in the nanofiber gas cell is 11-times larger. The paper is well written and the measurement matches very well with their simulation results and theoretical predictions.

In my opinion, the work is interesting for both the Brillouin integrated photonics community and the Brillouin microscopy and spectroscopy community. As for the former, the utilization of the evanescent optical field and the surrounding media acoustic modes might inspire novel designs of Brillouin integrated photonic devices. As for the latter, this work provides a method that doesn't require free-space light excitation. Overall, I would recommend this work to the publication in Nature Communications if the authors can address the following concerns:

1. In the second paragraph of the introduction, the authors claim that "all prior works were devoted to generating efficient Brillouin scattering in the core region of the waveguide. So far, no strong Brillouin scattering has been generated from an optical waveguide into the surrounding medium located in the waveguide's vicinity". However, in some works, for example the paper [1] that the authors also cited, researchers produce substantial Brillouin gain using dilute optical mode and the unguided, leaky

acoustic mode. The majority of the large spatial acoustic / optical overlap is in the SiO₂ cladding. Therefore, the authors need to further clarify their novelty on "evanescently-induced Brillouin scattering".

2. In Supplementary Section 4, the author performed a detailed simulation of Brillouin gain spectrum of a nanofiber immersed in water and predicted a high enough Brillouin signal to be observed. Is there any particular reason that the authors didn't conduct the measurement? It would be much more convincing if the authors could show some experimental data to confirm the feasibility of nanofiber waveguide for Brillouin spectroscopy and microscopy in liquid.

Some other minor points:

1. In figure 3, why is the measured 9 GHz signal from the silica nanofiber hybrid acoustic mode so larger than the simulation results?

2. In figure 4b) and figure 5, the authors showed skewed Brillouin Stokes gain spectra and explained it using a theory about the tapered transition region. Do anti-Stokes Brillouin Stokes gain spectra have the same asymmetry? Does it fit well with the theory, too?

3. The authors claim that the nanofiber gas cell can be used for pressure and temperature sensing. However, it is not clear in the paper how the authors can deal with the temperature and pressure cross-sensitivity.

4. The authors didn't mention their second half of the paper (different gas pressures and different gas types) either in the abstract or in the introduction.

[1]. Gundavarapu, S., Brodnik, G.M., Puckett, M. et al. Sub-hertz fundamental linewidth photonic integrated Brillouin laser. *Nature Photon* 13, 60–67 (2019).

Response to Reviewers

The authors are sincerely very grateful to the Reviewers for their time spent to evaluate our manuscript and their very valuable comments that will undoubtedly help to make our communication much better.

After receiving the reports, we thoroughly considered every question brought up by the reviewers and made the necessary revisions to improve the manuscript. In this response, we answer/address all concerns and comments in a point-to-point style. The reviewer's comments are in **Black**, our responses/answers are in **Blue**, and the actions/revisions we made for the revised manuscript are in **Red**.

Point-to-point response:

Reviewer #1 (Remarks to the Author):

Authors present a paper entitled “Large evanescently-induced Brillouin scattering at the surrounding of a nanofibre” in which the evanescent field in a tapered fibre is used to excite Brillouin scattering in a gas surrounding the waveguide. The manuscript is well-written and relatively easy to follow. It is definitely interesting. I would recommend publication after Authors consider the following:

1) This method is very much in the spirit of other evanescent wave sensors. Tapered fibres (or even those that have gone through the painful process of polishing into a D shape) and their resulting evanescent fields have been used for chemical, biological, etc. sensing using a wide range of modalities. How does the additional complexity of a Brillouin apparatus compare with some of those other methods? Can Authors propose advantages to such a sensor in using Brillouin scattering rather than some other framework?

Reply:

We are thankful to the Reviewer to raise this point. Tapered fibres and D-shape fibre with evanescent field have been used for chemical, biological and molecular detection based on absorption, fluorescence, elastic scattering and Raman scattering. So far, no waveguide-based evanescent Brillouin sensing has been reported. Our work provides an efficient way for Brillouin sensing and it is indeed in the spirit of evanescent wave sensors but based on different physics.

Nanofibre-based evanescent Brillouin sensing does not bring additional complexity compared with Raman or fluorescence-based spectroscopy. They all basically include a pump light and a spectral analyser for the backward scattering or fluorescent light detection. In our work, we used heterodyne detection for the spectral analysis.

Brillouin scattering measures the mechanical (or viscoelastic) properties of a material which absorption, fluorescence, elastic scattering or Raman scattering methods can't. Brillouin scattering has been widely used for spectroscopy (e.g. [Science Advances **6**:eabc1937 (2020)]) and microscopy (e.g. [Nature Methods **12**, 1132-1134 (2015)]). The sound wave velocity and the material viscosity can be obtained from the Brillouin resonant frequency and Brillouin linewidth

of the measured Brillouin spectrum, respectively [The Journal of Chemical Physics **150**, 154502 (2019); The Journal of Chemical Thermodynamics **30**, 1589-1601 (1998)]. Our results suggest a way to realise a novel waveguide-based Brillouin spectroscopy and microscopy that no longer requires free-space illumination which is required in the previous works.

Action taken:

In the discussion part of the revised manuscript, we have added the comparisons of our work with other evanescent field sensing [Optical and Quantum Electronics 26, S249-S259 (1994)] based on absorption [Optics Express 15, 11952-11958 (2007)], fluorescence [Optics Express 17, 21704-21711 (2009)], elastic scattering [Nature Photonics 11, 477-481 (2017)] and Raman scattering [Optica 6, 570-576 (2019)].

We have also added the discussion of the complexity of our method compared with Raman and fluorescence methods as well as the advantage of our sensor compared with other works in the discussion part of the revised manuscript.

2) *Authors state that “So far, no strong Brillouin scattering has been generated from an optical waveguide into the surrounding medium located in the waveguide’s vicinity.” The word “strong” is somewhat subjective and ambiguous. Can Authors provide examples where the evanescent (optical) wave was used to excite Brillouin scattering?*

Reply:

We thank the reviewer for pointing out our statement. We must admit that in the original manuscript, the statement “strong Brillouin scattering” without specifying previous evanescent Brillouin scattering is ambiguous. D. J. Blumenthal’s group has demonstrated evanescent Brillouin scattering using dilute optical mode and unguided acoustic mode in micro-resonator waveguides with Si₃N₄ core and silica cladding [Nature Photonics **13**, 60-67 (2019); Nature Communications **12**, 4685 (2021)]. This point has also been mentioned by the third reviewer. The single-pass Brillouin gain coefficients are 0.1 m⁻¹W⁻¹ at 1550 nm [Nature Photonics **13**, 60-67 (2019)] and 0.49 m⁻¹W⁻¹ at 674 nm [Nature Communications **12**, 4685 (2021)] respectively, and the substantial Brillouin gains result from the high-Q resonant enhancement. So far, no strong Brillouin scattering has been generated from a single-pass optical waveguide into the surrounding medium located in the waveguide’s vicinity.

Furthermore, our evanescent Brillouin gain coefficient is 18.9 m⁻¹W⁻¹ and the evanescent Brillouin scattering is in a fluid medium, which means it shows more flexibility for spectroscopy and microscopy compared with previous evanescent Brillouin works. It should be mentioned that our scheme can also be extended to high-Q suspended micro-resonator waveguides, which would unavoidably result in a much higher Brillouin scattering effect.

Action taken:

In the introduction part of the revised manuscript, we have specified the works on evanescent Brillouin scattering and added the reference. We have changed the sentence “So far, no strong Brillouin scattering has been generated from an optical waveguide into the surrounding medium located in the waveguide’s vicinity.” to “So far, no significant Brillouin scattering has been

generated from a single-pass optical waveguide into the surrounding medium located in the waveguide's vicinity." Also we have added "single-pass" in the abstract and discussion.

3) Authors consistently compare their results against those of a standard SMF, which has a very large mode size. A better comparison would be with fibres made from high-gain materials, such as high-index chalcogenides, or a microstructured highly nonlinear fibre. Looking at Fig. 4, even the "silica contribution" from the tapered fibre appears to have a higher gain coefficient than the conventional SMF.

Reply:

We fully agree that we must also compare the Brillouin gain with high-gain materials such as high-index chalcogenides and microstructured highly nonlinear fibres. Our Brillouin gain coefficient $18.9 \text{ m}^{-1}\text{W}^{-1}$ is 15-times higher compared to a small-core highly nonlinear microstructured fibre ($1.23 \text{ m}^{-1}\text{W}^{-1}$) [in Table 2 in Optics Letters **31**, 2541-2543 (2006)] and 8-times smaller compared to a high-index chalcogenide fibre ($153.8 \text{ m}^{-1}\text{W}^{-1}$) [calculated by dividing the peak Brillouin gain $6 \times 10^{-9} \text{ m/W}$ to the effective area 39 um^2 in Optics Express **13**, 10266-10271 (2005)].

Action taken:

In the introduction and discussion, we have added the comparison of our Brillouin gain with that of a small-core highly nonlinear microstructured fibre and a chalcogenide fibre.

Along those lines, is it possible to deduce a material value in units of m/W? The values for the gain coefficient provided are dependent upon the area of the optical mode. It would be helpful to the reader in understanding this in the context of the tradeoff between mode intensity and overlap with the outside region. This, too, is a place where information is lacking for the water simulation.

Reply:

Yes, we deduce the Brillouin gain coefficient (in units of m/W) of CO₂ using the Brillouin gain coefficient in gas-filled hollow-core fibre (in units of $\text{m}^{-1}\text{W}^{-1}$) and the acousto-optic overlap effective area in the Supplementary Table S3 in Nature Photonics **14**, 700-708 (2020). This is because the Brillouin scattering in hollow-core fibre is very similar to free-space condition. The Brillouin gain coefficient (in units of m/W) of 41 bar and 57 bar CO₂ at 1550 nm are estimated to be 1.34×10^{-10} and $2.60 \times 10^{-10} \text{ m/W}$.

The Brillouin gain coefficient (in units of m/W) of water at 780 nm was measured to be $5.96 \times 10^{-11} \text{ m/W}$ (from Supplementary Table 2 in Nature Methods **17**, 913-916 (2020)). Then we get the Brillouin gain coefficient of water to be $7.53 \times 10^{-11} \text{ m/W}$ at 694 nm by using the scattering gain-wavelength dependence relation (R. Boyd, Nonlinear Optics, Third edition (2008)).

Action taken:

We have added the gain coefficients in units of m/W of 57 bar CO₂ at 1550 nm and water at 694 nm in the discussion part of the revised manuscript and in Supplementary S5, respectively.

4) As far as applications go, Authors state that "This feature offers the attractive possibility to realise compact optical amplification stages, since a gain similar to that generated over several tens of meter of standard fibre can be obtained over a few centimetres along the nanofibre, with a

very minor insertion loss” and go on to state that this can be useful in photonic integrated circuits. There have been recent demonstrations of Brillouin amplification in, for example, silicon nanowaveguides where the high index gives rise to an even larger Brillouin gain. Can the proposed configuration outperform these other approaches? Even if so, do the proximity of the Brillouin frequency to the pump and the narrow width of the gain spectrum pose issues or obstacles for this application?

Reply:

In terms of gain coefficient or Brillouin amplification, our proposed configuration can't outperform silicon nanowaveguide because our gain coefficient is two orders of magnitude smaller compared with that of a silicon waveguide [Nature Photonics **10**, 463-467 (2016)]. The concept in our work can be applied to any waveguide with a substantial evanescent field such as suspended, uncladded or slot waveguide. Although the silicon waveguides have been designed for light-sound interaction, the interaction between the evanescent field of their guided light and the surrounding fluidic medium has not yet been exploited. Furthermore, our configuration allows tailoring the Brillouin gain spectrum by changing the gas pressure or using different type of gases.

But a key point is – if we consider that the light will come from a fibre and will end into a fibre like in most practical cases – is the quasi absence of insertion loss and the natural tapering, which is technologically way much simpler than engineering fibre-to-waveguide tapers in planar circuits.

In our experiments, we use backward Brillouin scattering and the pump and signal light counter-propagate. Therefore, in principle, the proximity of the pump and signal does not pose issues for light amplification unless there is reflection point in the fibre. The gain spectrum bandwidth of a nanofibre gas cell filled with 57 bar CO₂ is 15 MHz which is 2-fold narrower than a standard single-mode fibre. The peak gain coefficient of a nanofibre gas cell filled with 57 bar CO₂ is 79 times stronger than a standard single-mode fibre. We can modulate and engineer the pump spectrum to broaden the signal amplification spectrum because the amplification spectrum is the convolution of the pump light spectrum and the Brillouin gain spectrum.

Action taken:

In the discussion part of the revised manuscript, we have added the comparison of our Brillouin gain with that of a silicon waveguide as well as the emphasis of the gain spectrum tailoring capability and the extremely low insertion loss of our scheme. In the same paragraph, we have added the discussion on the impact of the proximity of pump and signal as well as the narrow linewidth of the gain spectrum for light amplification.

5) What was the purpose of adding the other gases since no numbers are given? Is a lower gain the cause of the noisy signal (I am assuming since the 5bar CO₂ case is shown, that is meant to be a comparison to a low-gain case)? What is the significance of the noise bandwidth?

Reply:

The purpose of adding the other gases is to show the possibilities of the evanescent Brillouin scattering of our nanofiber to be used for Brillouin spectroscopy and gas analysis as well as the flexibility of our platform in tailoring the Brillouin gain at will.

The noise defined by the standard deviation of the signal off-peak of the Brillouin spectrum, is determined by the shot noise of the input light to the photodetector, the electrical noise of the photodetector and electrical spectrum analyser (ESA) as well as the noise equivalent bandwidth of ESA. The input light power to the photodetector is dominated by the local oscillator which means the shot noise does not change with the gas pressure. In our case, the noise is not caused by a lower gain. 5 bar CO₂ and 57 bar CO₂ both have a noise level of $\sim 0.013 \text{ m}^{-1}\text{W}^{-1}$.

In Fig. 6, the noise for 2 bar SF₆, 5 bar CO₂ and 10 bar N₂ are $0.0046 \text{ m}^{-1}\text{W}^{-1}$, $0.013 \text{ m}^{-1}\text{W}^{-1}$ and $0.015 \text{ m}^{-1}\text{W}^{-1}$, respectively. 5 bar CO₂ and 10 bar N₂ have similar noise level because they have the same ESA noise equivalent bandwidth. 2 bar SF₆ has ~ 3 times smaller noise level because it has 10 times smaller noise equivalent bandwidth compared with 5 bar CO₂ and 10 bar N₂. The SNR for 2 bar SF₆, 5 bar CO₂ and 10 bar N₂ are 28, 26 and 21 respectively. The results show our configuration has substantial SNR even with low gas pressure.

Action taken:

In the “Different types of gas” part of the revised manuscript, we have added the peak gain coefficients and the noise level as well as the signal-to-noise ratio for the gases shown in Fig. 6.

6) Section S2: This discussion is somewhat on the light side and written as an afterthought. More detail would be useful. It would also be informative/illustrative to show a few Brillouin spectra together from different positions along the taper. I would have imagined that the contributions from the tapered regions, particularly their tails, would have impacted the red side of the spectra, as well as the blue, but not quite as much.

Reply:

Fig. R1. Theoretical Brillouin spectra in 40 bar CO₂ obtained in a nanofibre with a waist of 740 nm. The red line shows the theoretical Brillouin spectrum at the uniform nanofibre waist region; the blue line shows the tapered regions; the black line shows the total Brillouin spectrum.

Fig. R2. Effective refractive index as a function of rod diameter.

Fig. R1 shows the different contributions of a nanofibre on the Brillouin spectrum. The red and blue lines are the theoretical Brillouin spectrum at the uniform nanofibre waist region and tapered region, respectively. The black line is the total Brillouin spectrum.

Fig. R2 illustrates the effective refractive index as a function of the rod diameter. The resonant Brillouin frequency shift $\nu_B = 2n_{eff} V_a/\lambda$, where n_{eff} is the effective refractive index of the optical mode, V_a is the acoustic velocity of the surrounding gas and λ is the pump wavelength in vacuum. Since the tapered regions have larger diameter than the waist region, the effective refractive index in the tapered regions is larger than in the waist region. Therefore, the Brillouin frequency shift in the tapered regions is larger than in the waist region which is shown in Fig. R1.

Action taken:

In the revised Supplementary S3, we have added these two figures to illustrate the Brillouin spectrum from different regions and explain why the tapered region contribute to the higher frequency region of the Brillouin spectrum.

Therefore, I wonder about the fittings used for the spectral width (half width on the left side, as Authors state, such as the blue curve in Fig. 5(b)). Perhaps this method may be overestimating the spectral width?

Reply:

Fig. R3. Simulation results with different tapered profile. The rod diameter (a) and the Brillouin frequency shift (b) as a function of the elongation for three tapered profiles. (c) The normalised Brillouin spectra for three tapered profiles. Solid lines show the total Brillouin spectra and dotted lines show the contribution from the tapered regions for three tapered profiles.

We calculate the Brillouin spectrum for different taper profile. As we can see in Fig. R3(c) the Brillouin spectrum is not affected by the taper profile because the optical evanescent field is really small for diameter larger than 1 μ m. The tapered profile has little impact on the left side of the Brillouin gain spectrum, therefore using the fitting from the left side does not overestimate the spectral width.

Action taken:

In the revised Supplementary S3, we have added Fig. R3 to explain that the tapered profile has little impact on the left side of the Brillouin gain spectrum.

Reviewer #2 (Remarks to the Author):

In the manuscript, Yang et al. report Brillouin scattering in the evanescent field of a sub-wavelength waveguide in a high-pressure gas cell. The experiment results show clear Brillouin scattering from the evanescent field and are in good agreement with the simulation results. Results using different gas pressures and different gas species are compared to show the potential for Brillouin spectroscopy. The manuscript is well written.

However, one of the main claims the authors made is that the Brillouin gain coefficient is much larger than in their previous results and SM fiber in particular. While these statements are correct, it does not seem a fair comparison, as the current gas pressure used is 40 bar which is a very high number. Also, as stated by the authors, this pressure is already approaching the gas-liquid phase transition. For some of the lower pressures measured (e.g., the 5 bar trace in Fig. 5) the Brillouin scattering peak is hardly resolved. Even though the results are interesting and clearly presented, the usefulness and generality of the approach as a spectroscopic technique or new platform for various studies as mentioned in the abstract is in question. It seems that the results are more appropriately presented in a journal having a focused subject area.

Reply:

We understand the point raised by the reviewer, but spectroscopy is just one aspect of the application field of our proposition and large photonic amplification is another important field, which potential deserves to be illustrated with high pressure gas. We did not want to multiply the illustrations in the paper, but we can easily demonstrate hereafter that the signal is actually important enough for spectroscopy under ambient conditions.

The Brillouin spectrum for lower pressures CO₂ (e.g. the 5 bar trace in Fig. 5) is hardly resolved in Fig. 5 because it is illustrated with 57 bar in the same figure. In the original manuscript, the 5 bar CO₂ trace was zoomed-in and shown in Fig. 6. In Fig. 6, the noise, defined by the standard deviation of the off-peak of the Brillouin spectrum, for 2 bar SF₆, 5 bar CO₂ and 10 bar N₂ are 0.0046 m⁻¹W⁻¹, 0.013 m⁻¹W⁻¹ and 0.015 m⁻¹W⁻¹ respectively. 5 bar CO₂ and 10 bar N₂ have similar noise level because they have the same electrical spectrum analyser (ESA) noise equivalent bandwidth. 2 bar SF₆ has ~3 times smaller noise level because it has 10 times smaller noise equivalent bandwidth compared with 5 bar CO₂ and 10 bar N₂. The SNR for 2 bar SF₆, 5 bar CO₂ and 10 bar N₂ are 28, 26 and 21 respectively. The results clearly show the possibilities of the evanescent Brillouin scattering of our nanofibre to be used for Brillouin spectroscopy and gas analysis, even at or close to ambient conditions.

Furthermore, we have simulated the Brillouin spectrum for water and the water Brillouin gain coefficient is $1 \text{ m}^{-1}\text{W}^{-1}$ which is 8 times larger than 2 bar SF_6 . We believe the evanescent Brillouin scattering of our nanofibre can also be used for Brillouin microscopy and liquid analysis.

Action taken:

In the “Different types of gas” part of the revised manuscript, we have added the peak Brillouin coefficient, the noise level as well as the SNR for 2 bar SF_6 , 5 bar CO_2 and 10 bar N_2 . We have added a sentence saying that the signal is important enough under ambient conditions for a proper spectroscopic application.

The followings are some detailed points:

1. In Fig. 1a, the authors should provide discussion as to why Stokes and anti-Stokes are both favorable in the experiment?

Reply:

Regarding the low frequency of phonons and the large number that are thermally activated at ambient temperature, the creation (by a Stokes process) or the annihilation (by an anti-Stokes process) of one phonon does not perceptibly modify the total number of phonons, so that the scattering cross-section for spontaneous Brillouin scattering is identical for Stokes and anti-Stokes processes. It means they can be equally used for a similar result.

Action taken:

In the operation scheme part, we have added the discussion: “In spontaneous Brillouin scattering, both Stokes and anti-Stokes scattering are favorable for measuring the Brillouin spectrum [Physical Review A 42, 5514 (1990)]” and “For the anti-Stokes process, an incident pump photon is converted to a higher frequency anti-Stokes photon through the scattering from the acoustic wave and thereby annihilating a phonon, but this does not impact perceptibly on the total number of phonons that are thermally activated regarding their low frequency. In this work, we arbitrarily use Stokes scattering to measure the Brillouin spectrum.”

2. In Fig. 1c, it seems like the authors want to demonstrate the acoustic field in the Brillouin process. Is this elastic wave entirely composed of CO_2 , and confined to the waveguide boundary or is it a hybrid wave?

Reply:

Fig. R4. Numerical simulations of Brillouin scattering in nanofibre surrounding by 40 bar CO₂. (a) Optical power density of the fundamental TE-like mode for pump wavelength of 1550 nm. Spatial distribution of axial (b) and transverse (c) displacement, respectively. (d) Spatial distribution of elastic energy density of resonant phonon in CO₂ at 350 MHz.

The elastic wave generated by the optical evanescent field in the CO₂ around the nanofibre is confined to the boundary of the nanofibre. As we can see in Fig. R4, the most important contribution of the elastic wave comes from longitudinal displacement. Fig. R4 shows the transverse displacement has negligible contribution to the elastic wave, so that we can confidently claim that it is predominantly present in CO₂.

Action taken:

We have added the simulations of the transvers displacement and the elastic wave density in Supplementary S1.

3. In Fig. 3, there are surface acoustic modes at 5-6 GHz (about three times smaller than 4. GHz). Can this feature still be resolved in Fig. 4?

Reply:

Generally, surface acoustic wave (SAW) is very sensitive to the environment and can be used as mass or gas sensor [Sensors and Actuators B 8, 33-40 (1992)]. However, for high pressure gas, the SAW is strongly attenuated and cannot be resolved in Fig. 4. The gas surrounding the nanofibre plays the role of an acoustic damper.

5. In Fig. 4, the authors state that "The right peaks identify the Stokes scattering while the left peaks originate from the anti-Stokes scattering." As I understand, AOM will down-shift the LO frequency. Thus the left peak should also come from the down-shifted wave, which is the Stokes wave.

Reply:

AOM can down-shift or up-shift the LO frequency. For example: <https://www.brimrose.com/fiber-coupled-ao/fiber-coupled-frequency-shifters>

In our experiment, the AOM up-shifts the LO by +110 MHz which has been described in the original manuscript Fig. 4 caption. So the right peaks identify the Stokes scattering while the left peaks originate from the anti-Stokes scattering. Thank you for offering the opportunity to clarify this point that may rightfully raise some questioning.

6. In Page 5, the gain coefficient for SMF should also be given with error bar.

Reply and Action taken:

We thanks the reviewer for the comment. We have added the error bar for SMF in the revised manuscript.

7. In Fig. 6, why do the different lines have different RBW?

Reply:

Before the experiment, we theoretically estimated that the Brillouin gain coefficient of 2 bar SF₆ is several times smaller compared with 5 bar CO₂ and 10 bar N₂. Smaller RBW means more integration (average) time and better SNR. In order to achieve similar SNR for SF₆, CO₂ and N₂, we selected 10 times smaller noise equivalent bandwidth for SF₆ in contrast with CO₂ and N₂.

Action taken:

In the “Different types of gas” part of the revised manuscript, we have added the Brillouin coefficient, the noise and the SNR for 2 bar SF₆, 5 bar CO₂ and 10 bar N₂.

Reviewer #3 (Remarks to the Author):

In this paper, Fan et al. claim to demonstrate strong Brillouin scattering generated by the evanescent field of a sub-wavelength nanofiber waveguide in a pressurized gas cell. Compared with the authors' previous work of large Brillouin amplification in a gas-filled hollow-core-fiber at high pressure, the maximum gain coefficient achieved in the nanofiber gas cell is 11-times larger. The paper is well written and the measurement matches very well with their simulation results and theoretical predictions.

In my opinion, the work is interesting for both the Brillouin integrated photonics community and the Brillouin microscopy and spectroscopy community. As for the former, the utilization of the evanescent optical field and the surrounding media acoustic modes might inspire novel designs of Brillouin integrated photonic devices. As for the latter, this work provides a method that doesn't require free-space light excitation. Overall, I would recommend this work to the publication in Nature Communications if the authors can address the following concerns:

1. In the second paragraph of the introduction, the authors claim that “all prior works were devoted to generating efficient Brillouin scattering in the core region of the waveguide. So far, no strong Brillouin scattering has been generated from an optical waveguide into the surrounding medium located in the waveguide's vicinity”. However, in some works, for example the paper [1] that the authors also cited, researchers produce substantial Brillouin gain using dilute optical mode and the unguided, leaky acoustic mode. The majority of the large spatial acoustic / optical overlap is in the SiO₂ cladding. Therefore, the authors need to further clarify their novelty on “evanescently-induced Brillouin scattering”.

Reply:

We thank the reviewer for pointing out a potential controversy with our claim. Our previous claim is actually outdated since D. J. Blumenthal's group has demonstrated evanescent Brillouin scattering using dilute optical mode and unguided acoustic mode in micro-resonator waveguides with Si₃N₄ core and silica cladding [Nature Photonics **13**, 60-67 (2019); Nature Communications **12**, 4685 (2021)]. In these two works, the single-pass Brillouin gain are 0.1 m⁻¹W⁻¹ and 0.49 m⁻¹W⁻¹ which are much smaller than in our work. The substantial Brillouin gain results from the high-Q enhancement.

Action taken:

In the second paragraph of the introduction, we have rephrased “all prior works were devoted to generating efficient Brillouin scattering in the core region of the waveguide. So far, no strong Brillouin scattering has been generated from an optical waveguide into the surrounding medium located in the waveguide's vicinity” to “It should be mentioned that ultra-narrow linewidth Brillouin lasers based on evanescent Brillouin scattering using dilute optical mode and unguided leaky acoustic mode, have recently been demonstrated in micro-resonator waveguides with Si₃N₄ core and silica cladding. The single-pass Brillouin gain coefficients are 0.1 m⁻¹W⁻¹ at 1550 nm [Nature Photonics **13**, 60 (2019)] and 0.49 m⁻¹W⁻¹ at 674 nm [Nature Communications **12**, 4685 (2021)] respectively, and the substantial Brillouin gains result from the high-Q resonant enhancement. So far, no significant Brillouin scattering has been generated from a single-pass optical waveguide into the surrounding medium located in the waveguide's vicinity.”

We have also added “single-pass” in the abstract and discussion of the revised manuscript.

2. In Supplementary Section 4, the author performed a detailed simulation of Brillouin gain spectrum of a nanofiber immersed in water and predicted a high enough Brillouin signal to be observed. Is there any particular reason that the authors didn't conduct the measurement? It would be much more convincing if the authors could show some experimental data to confirm the feasibility of nanofiber waveguide for Brillouin spectroscopy and microscopy in liquid.

Reply:

[redacted]

Fig. R5. Absorption spectrum of liquid water from Wikipedia. The data is from Applied Optics 32, 3531-3540 (1993).

In our experiment, all the system worked at 1550 nm band. Fig. R5 shows the water absorption coefficient at 1550 nm band is more than 3 orders of magnitude higher than in the visible band. To do water Brillouin experiment, visible band must be selected. We can't conduct the water experiment because we don't have laser and optical components and devices at visible wavelength. This experiment would go beyond the scope of the present study, regarding the change in the equipment to the visible range and the special handling of a high purity liquid. But this is definitely something we have in mind for future prospects.

We have already simulated the Brillouin scattering in a nanofibre surrounded with water. In order to make our calculation more convincing, here we will do an intuitive estimation using the measured results by previous works. The measured Brillouin coefficient of water at 780 nm in units of m/W is 5.9×10^{-11} m/W [Nature Methods 17, 913-916 (2020)]. The measured Brillouin coefficient of 40 bar CO₂ at 1550 nm is 1.34×10^{-10} m/W [Nature Photonics 14, 700-708 (2020)]. In our work, the measured Brillouin gain of a nanofibre filled with 40 bar CO₂ (in units of m⁻¹W⁻¹) at 1550 nm is 8.2 m⁻¹W⁻¹. So we estimated the Brillouin coefficient of water (in units of m⁻¹W⁻¹) to be 3.6 m⁻¹W⁻¹. This estimated value is 3.6 times larger than our simulated value which may result from the larger mode area in a nanofibre surrounded with water compared to a nanofibre surrounded with 40 bar CO₂ gas.

Action taken:

In Supplementary S5, we have added the intuitive estimation from previous measured results.

Some other minor points:

1. In figure 3, why is the measured 9 GHz signal from the silica nanofiber hybrid acoustic mode so larger than the simulation results?

Reply:

The blue and dotted red lines in Fig. 3 are respectively the experimental and numerical simulation results in the nanofibre after the tapering process. The dotted black line in Fig. 3 is the measured Brillouin spectrum after applying the strain. As a result of the strain, the resonances due to surface acoustic modes and hybrid acoustic modes both shift to higher frequencies, perfectly matching with our previous analysis [APL Photonics 4, 080804 (2019)]. Specifically, the resonant frequency of the main peak of a hybrid acoustic mode shift from 8.2 GHz to 9 GHz. We explained this in the original manuscript.

In Fig. 3, the amplitude of the measured hybrid acoustic mode of silica nanofibre with a strain of $\sim 2.15\%$ at 9 GHz is 48. The measured resonant Brillouin frequency of the same hybrid acoustic mode before applying a strain is at 8.2 GHz and the amplitude is 63. The resonant Brillouin frequency of the simulation result before applying strain is at 8.2 GHz with an amplitude of 38. The measured Brillouin signal due to the hybrid acoustic mode is very close to the simulation result.

Action taken:

In the second paragraph of the “Nanofibre gas cell” section in the revised manuscript, we have specified the hybrid acoustic mode before and after applying the strain.

2. In figure 4b) and figure 5, the authors showed skewed Brillouin Stokes gain spectra and explained it using a theory about the tapered transition region. Do anti-Stokes Brillouin Stokes gain spectra have the same asymmetry? Does it fit well with the theory, too?

Reply:
Fig. R6. Experimental Brillouin spectra for the Stokes and anti-Stokes Brillouin scattering.

In theory, the anti-Stokes gain spectrum is the same as the Stokes gain spectrum in spontaneous Brillouin scattering [Physical Review A 42, 5514-5521 (1990)]. Fig. R6 shows that the measured anti-Stokes and Stokes scattering of the nanofibre filled with 40 bar CO₂ have the same Brillouin spectrum. So it also fits well with the theory.

Action taken:

In the first paragraph of the “Different gas pressures” section in the revised manuscript, we have added “Both Stokes and anti-Stokes Brillouin scattering have exactly the same spectrum.”.

3. *The authors claim that the nanofiber gas cell can be used for pressure and temperature sensing. However, it is not clear in the paper how the authors can deal with the temperature and pressure cross-sensitivity.*

Reply:

We thank the reviewer for pointing out the cross-sensitivity. The reviewer is definitely right. Our nanofibre gas cell cannot be used for simultaneous temperature and pressure sensing. It can only be used for pressure or temperature sensing, unless another feature of the Brillouin spectrum showing different temperature and pressure dependence can be efficiently exploited (e.g. linewidth). This is because the temperature and the pressure will make the spectrum shift, but the pressure will moreover modify the Brillouin linewidth.

Action taken:

In the third paragraph of the Discussion part, we have changed “Our nanofibre gas cell can also be used for novel approaches in pressure and temperature sensing.” To “Our nanofibre gas cell can also be used for novel approaches in pressure or temperature sensing.” .

4. *The authors didn't mention their second half of the paper (different gas pressures and different gas types) either in the abstract or in the introduction.*

Reply and Action taken:

We thank the reviewer for pointing out this. We have added two sentences in the last paragraph of the introduction: “We observe drastic Brillouin scattering enhancement by increasing the gas pressure.” and “We also measure the Brillouin spectra for different types of gases such as carbon dioxide (CO₂), sulfur hexafluoride (SF₆) and nitrogen (N₂).”

[1]. Gundavarapu, S., Brodrik, G.M., Puckett, M. et al. Sub-hertz fundamental linewidth photonic integrated Brillouin laser. *Nature Photon* 13, 60–67 (2019).

REVIEWER COMMENTS

Reviewer #1 (Remarks to the Author):

Authors have carefully and thoughtfully responded to my questions and comments and have made the appropriate changes. I recommend publishing the paper.

Reviewer #2 (Remarks to the Author):

The authors have adequately addressed by previous comments and the paper is acceptable for publication.

Reviewer #3 (Remarks to the Author):

The reviewer would like to thank the authors for addressing all the comments from the previous review. I am satisfied with their answers and have no more comments to add. From my point of review, the paper is suitable for publication.